# Relationship between the composition of the intestinal microbiota and the tracheal and intestinal colonization by opportunistic pathogens in intensive care patients

**Candice Fontaine[1], Laurence Armand-Lefèvre[1,2], Mélanie Magnan[1], Anissa Nazimoudine[2], Jean-François Timsit[1,3], Etienne Ruppé[1,2]***

1 Université de Paris, IAME, INSERM, Paris, France, 2 AP-HP, Laboratoire de Bactériologie, Hôpital Bichat, Paris, France, 3 AP-HP, Service de Réanimation Médicale et Infectieuse, Hôpital Bichat, Paris, France

* etienne.ruppe@inserm.fr

## Abstract

### Objective

Infections caused by multidrug-resistant Gram-negative bacilli (MDR-GNB) are a major issue in intensive care. The intestinal and oropharyngeal microbiota being the reservoir of MDR-GNB. Our main objective was to assess the link between the composition of the intestinal microbiota and the tracheal and intestinal colonization by MDR-GNB, and also by *Enterococcus* spp. and yeasts.

### Methods

We performed a 2-month prospective, monocentric cohort study in the medical intensive care unit of our hospital. Patients ventilated >3 days and spontaneously passing feces were included. A fecal sample and an endotracheal aspiration (EA) were collected twice a week. MDR-GNB but also *Enterococcus faecium* and yeasts (as potential dysbiosis surrogate markers) were detected by culture methods. The composition of the intestinal microbiota was assessed by 16S profiling.

### Results

We collected 62 couples of feces and EA from 31 patients, including 18 feces and 9 EA positive for MDR-GNB. Forty-eight fecal samples were considered for 16S profiling. We did not observe a link between the diversity and the richness of the intestinal microbiota and the MDR-GNB intestinal relative abundance (RA). Conversely, we observed a negative link between the intestinal diversity and richness and the RA of *Enterococcus* spp. (p<0.001).

### Conclusion

The fecal MDR-GNB RA was not associated to the diversity nor the richness of the intestinal microbiota, but that of *Enterococcus* spp. was.

**Data Availability Statement:** The reads have been deposited at the NCBI SRA (access number PRJNA641109).

**Funding:** This work was partially supported by the "Fondation pour la Recherche Médicale" (Equipe FRM 2016, grant number DEQ20161136698)." The rest of the funding was internal to our UMR1137 IAME research unit (funding from INSERM and University of Paris).

**Competing interests:** The authors have declared that no competing interests exist.

## Introduction

The increased prevalence of multidrug-resistant Gram-negative Bacilli (MDR-GNB) in both community and healthcare-acquired infections is a major public health issue. Especially, resistance to third generation cephalosporins via the production of extended-spectrum beta-lactamases (ESBL) and/or AmpC-type cephalosporinases in GNB have put carbapenems as the drugs of choice in the treatment of these MDR-GNB infections. The increase in their consumption has in turn led to the emergence and carbapenem-resistant GNB such as Enterobacterales (especially those producing carbapenemases), and *Pseudomonas aeruginosa* which are considered as the most serious threats to global health by the World Health Organization (WHO, www.who.int/drug). Hence, containment strategies to cope with MDR-GNB are urgently expected.

Humans are colonized by a number of microorganisms (mostly bacteria) approximatively equivalent to the number of our own cells [1], the largest part being located in our digestive tract. The intestinal microbiota indeed harbors an estimated several hundred different species among which anaerobic bacteria (mainly Firmicutes and Bacteroidetes) are dominant with $10^{12}$ to $10^{14}$ colony-forming units [CFU] per gram of feces [2]. It also consists of potentially pathogenic bacteria (referred to as opportunistic pathogens) that are subdominant, including Enterobacterales and enterococci (approximately $10^8$ CFU per gram of feces) [3]. One of the main roles of the intestinal microbiota is to exert a barrier effect against pathogenic bacteria through a mechanism called resistance to colonization [4]. Indeed, some dominant anaerobic bacteria oppose to the sustained colonization by exogenous bacteria, including MDR-GNB.

The gastrointestinal tract is the primary reservoir for the bacterial pathogens that cause most nosocomial infections [5]. Indeed, antibiotics alter the intestinal microbiota by eliminating the susceptible bacteria (including those exerting the resistance to colonization) and promoting the overgrowth of antibiotic-resistant microorganisms [6]. The multiplication of antibiotic-resistant bacteria in the microbiota leads to an increased risk that they could be involved in subsequent infections such as digestive translocation [7] and urinary-tract infections [8]. Besides the intestinal microbiota, the oropharyngeal MDR-GNB colonization is known to significantly increase the probability of finding these bacteria in the respiratory tract and therefore the risk of MDR-GNB ventilator associated pneumoniae (VAP) [9]. Intestinal and oropharyngeal colonization seem closely linked and a MDR-GNB primarily found in the gut can be later found in the oropharynx during prolonged hospitalization [10]. Moreover, the oropharyngeal carriage of one given bacterium significantly increases the probability of it being found in respiratory samples in case of infection [11]. The composition of the oropharyngeal microbiota could also play a role in the interplay between pathogen colonization and ventilator-associated pneumonia. Indeed at the time of intubation, the oropharyngeal microbiota of patients subsequently developing VAP was found to differ from that of patients not developing VAP [12].

Intensive care unit (ICU) patients are particularly vulnerable to infections. During their stay in the ICU, approximately 20% patients develop a hospital-acquired infection [13] which represents the leading cause of death [14]. Most of these infections are ventilator associated pneumonia (VAP), one third of which are caused by MDR-GNB [15]. In total, MDR-GNB are responsible of more than 40% of hospital-acquired infections in ICU and double the relative risk of death [16]. In 50 to 80% of cases, ICU patients receive a wide array of antibiotics aiming at being active on potentially resistant pathogens. Indeed, early adequate antibiotic therapy reduces the mortality by more than 10% together with a global reduction duration of the stay. Over the last few years, the intestinal microbiota of ICU patients has been a matter of interest. Early observations showed that it was indeed under major changes during the stay likely due

to the various medication (including antibiotics) the patients received [17, 18]. In many cases, the richness (i.e. the number of unique bacteria) and the diversity (i.e. the balance of their distribution) of the intestinal microbiota significantly dropped and some bacterial species such as *Enterococcus* spp. or yeasts such as *Candida* spp. ended up being the dominant microorganisms in the gut [19, 20]. Furthermore, the extent of the alteration of the microbiota could be linked to mortality. At admission to the ICU, Freedberg et al. observed that patients who had an intestinal dominance (i.e. >30% of all reads) of *Enterococcus* spp. had a worse outcome than those who had not [21] Agudelo-Ochoa et al. found that a high abundance of Enterococci in the gut of septic ICU patients was associated to death [22]. Besides, the bacterial diversity of the respiratory tract was observed to be negatively correlated to the mortality [23]. Yet the link between the global composition of the microbiota and specific bacteria such as MDR-GNB, Enterococci and yeasts has not been assessed, which is what we aim to do in the present work.

## Material and methods

This is a monocentric prospective cohort study performed in the 36-bed medical intensive care unit (ICU) of the 900-bed Bichat-Claude Bernard university teaching hospital in Paris, France.

### Population

Adult patients were considered for the study if they were admitted to the medical ICU of our hospital between January 1 and March 1, 2018. All patients with mechanical ventilation at admission with a predicted duration of ventilation longer than three days and spontaneously passing feces were included in the study. The study was approved by the ethics review committee for biomedical research projects, Paris Nord (authorization number 2018–005). According to the French regulation, the patient was informed while the need for signed consent was waived.

### Sample collection

For each patient, the first feces passed after admission was collected and then twice a week when possible (when a stool was passed) until discharge or death. At the time the feces were collected, an endotracheal aspiration (EA) was also performed. All fecal samples were spontaneously passed. EA were collected by the nurse or the investigator (CF) with an aspiration in the intubation probe. Samples were kept at +4˚C in the ICU before being sent to the bacteriology laboratory of the Bichat-Claude Bernard Hospital. Approximately 100 mg of the feces and 100 μl of the EA samples were frozen at -80 degrees (one 30% glycerol brain heart infusion (BHI) tube and one tube without preservative).

### Sample culture

In this study, MDR-GNB referred to as (1) extended-spectrum beta-lactamase (ESBL), carbapenemase and/or high-level AmpC producing Enterobacterales and, (2) ceftazidime-resistant *P. aeruginosa* and (3) *Stenotrophomonas maltophilia*. For culturing purposes, approximately 100 mg of feces were diluted in 10 mL of 0.9% sodium chloride while EA were used without dilution. MDR-GNB were searched by culturing 100 μL the feces dilution or the EA on selective media: ChromID® ESBL media (bioMérieux, Marcy-l'Etoile, France), 1 mg/L cefotaxime supplemented Drigalski agar plates (Bio-Rad, Marne-la-Coquette, France) and Cetrimide media on which was deposited a disc of ceftazidime in the 2nd quadrant (bioMérieux). Besides, we also searched for *Enterococcus faecium* by using a Columbia colistin nalidixic agar media

(bioMérieux) on which was deposited a disc of imipenem in the $2^{nd}$ quadrant. For yeast, we used the ChromID$^{®}$ Candida media (BioMérieux). All plates were incubated at 35°C under aerobic conditions for 24h. All colony-forming units (CFUs) that had grown on selective media were identified by matrix-assisted laser desorption ionization-time (MALDI-TOF) mass spectrometry (Bruker, Bremen, Germany) and then tested for antibiotics susceptibility by the disc diffusion method, according to the EUCAST 2018 v1 recommendations that applied at the time of the study. No antimicrobial susceptibility testing was performed for *E. faecium* and yeasts. For MDR-GNB positive samples, concentration of total aerobic GNB and of MDR-GNB were determined by plating serial dilutions (pure, $10^{-2}$, $10^{-4}$) of initial feces or EA sample onto Drigalski agar (Bio-Rad) with or without 1mg/L cefotaxime. After 24h of incubation at 35°C, CFU were counted in decimal logarithms at the dilution in which 10 to 100 CFU grew (CFU per gram of feces and CFU per millilitre). MDR-GNB relative abundance (MDR-GNB RA) was calculated as the ratio of MDR-GNB concentrations divided by the total number of aerobic GNB, expressed as a percentage or in log 10 [8]. For patients who carried more than one MDR-GNB bacteria in feces or EA, the MDR-GNB RA was the RA of the total MDR-GNB.

### DNA extraction 16S RNA gene sequencing

All frozen fecal samples were thawed, and total DNA extraction was performed using the QiAamp DNA stool Mini Kit (Qiagen, Courtaboeuf, France) according to the local protocol used in our laboratory (see supplementary material. The DNA was measured before freezing at -20°C using the Qubit$^{®}$ instrument (ThermoFisher Scientific, Montigny-le-Bretonneux).

### 16S RNA gene sequencing

The V4 hypervariable region of the 16S ribosomal RNA gene was amplified and sequenced were sequenced for using the Illumina MiSeq platform. The protocol used followed the 16S metagenomic sequencing library protocol (15044223B) provided by Illumina. This protocol created a final amplicon of 428 base pairs spanning to the V4 region. All the amplicons from fecal samples with satisfactory acceptance criteria for sequencing (a quantitative DNA assay coding for the V4 region of 16S RNA was performed using the Qubit$^{®}$ instrument and the quality of the amplification was assessed by migrating the DNA from each fecal sample in an agarose gel) were finally considered for sequencing.

### Bioinformatic analyses

The analysis of 16S rRNA sequences were performed using Shaman (Shiny Application for Metagenomic Analysis, http://shaman.pasteur.fr/). Reads processing and relative abundance counts were based on a negative binomial regression (deseq2 R package) [24]. Operational taxonomic units (OTUs) were clustered at 97% sequence similarity. Diversity assessed by the Shannon index and richness was calculated at the operational taxonomic unit (OTU) level.

### Statistical analysis

Statistical analyses were performed using R v3.4.2 (package R, deseq2). Population characteristics were expressed as medians and percentages. The quantitative variables of absolute concentrations (CFU per gram of feces or CFU per millilitre) and relative abundance (in log 10 or percentage) were expressed as means and medians (minimum and maximum values). The statistical tests used were Student t test and Pearson tests. The significance level was set at a value

of 0.05. Figures were designed using ggplot2 and colours were choose using colorbrewer2 (https://colorbrewer2.org).

# Results

## Population

We included 31 patients who were able to emit feces spontaneously the day of inclusion. Characteristics of patients are showed in the Table 1 and detailed in the S1 Table. They were predominantly male (65%), with a median age of 59 years [range 22; 75], with high severity scores at admission (SOFA 8 [range 1; 16] and SAPS II 52 [range 25; 106]) and 42% were immunocompromised. The median length of hospitalization of the 31 patients was 17 days. We collected a total of 124 samples (62 feces and 62 EA) from these 31 patients over 2 months (Fig 1). For each patient, we collected an average of 2 [range 1; 8] fecal and EA samples.

## Culture results

Twenty-seven MDR-GNB positive samples were obtained from 13 patients. As for the feces, 29% (18/62) were positive for MDR-GNB: ESBL-producing *E. coli* (n = 9), ESBL-producing *Klebsiella pneumoniae* (n = 4), ESBL-producing *Citrobacter freundii* (n = 1), AmpC-overproducing *Hafnia alvei* (n = 1, AmpC-overproducing *Morganella morganii* (n = 1), AmpC-overproducing *Enterobacter aerogenes* (n = 1), AmpC-overproducing *Enterobacter cloacae* (n = 1) and *Stenotrophomonas maltophilia* (n = 1). The mean relative abundance was 57.0 percent and the mean absolute intestinal concentration of MDR-GNB was 8.2 CFU per gram of feces [4.5–10.5].

Fourteen percent of EA (9/62) were positive for MDR-GNB: *S maltophilia* (n = 4), ESBL-producing *K pneumoniae* (n = 2), ESBL-producing *E coli* (n = 1), ESBL-producing *C freundii* (n = 1) and AmpC-overproducing *H alvei* (n = 1). The mean relative abundance of MDR-GNB in EA was 90%, it was significantly higher than in the gut (Student test, p = 0.03, S1 Fig in S1 File). Five samples were positive for MDR-GNB in both feces and EA in 4 different patients. Twenty-two samples were positive for *E. faecium* including 13 fecal samples (21%) and 9 EA (15%). Forty-one samples were positive for at least one yeast, including 27 fecal samples (44%) and 14 EA (22%).

## Sequencing results

Among the 62 fecal samples, 48 were submitted to 16S profiling (S2 Table). We considered all the samples positive for MDR-GNB (n = 18) and samples with *E. faecium* and/or a yeast as detected in culture while no MDR-GNB was cultured (n = 11). We also considered samples from patients with an infection caused by an MDR-GNB (despite EA and feces were negative for MDR-GNB, n = 1), 14 samples with negative cultures but obtained from patients who had other positive samples to MDR-GNB, *E. faecium* and/or yeast during the follow-up and 4 samples from a patient with no positive culture at any time. The average number of reads obtained after sequencing was 325,500 (median 237,700 [1,406; 1,175,604]). After quality filtering, the average length of the reads obtained on the forward strand was 246 bases. On the reverse strand, the average length of the reads after quality filtering was 238 bases. The average number of combined pairs obtained was 210,400 and the average mapping rate was 97%.

## MDR-GNB intestinal colonization and composition of intestinal microbiota

We did not observe a difference between the diversity (Student test, p = 0.4) and richness (Student test, p = 0.84) of the intestinal microbiota according to the MDR-GNB intestinal colonization (Fig 2 panels A and B). Nor did we observe a link between the diversity (Pearson

**Table 1. Characteristics of the patients (n = 31) included in the study.**

| Clinical characteristics of patients at inclusion (n = 31) | Values (%) or median (min-max) |
|---|---|
| **Gender** | |
| Male | 20 (64.5%) |
| Female | 11 (35.5%) |
| **Age (years)** | 59 (22–75) |
| **Background information** | |
| Neoplasia | 2 (6.5%) |
| Organ transplantation | 5 (16.1%) |
| Autoimmune disease | 3 (9.7%) |
| Asplenia | 0 (0%) |
| HIV | 2 (6.5%) |
| **Immunosuppressive treatments** | |
| None | 20 (64.5%) |
| Corticosteroids | 6 (19.4%) |
| Immunosuppressants | 7 (22.6%) |
| Immunoglobulins | 2 (6.5%) |
| Antiretrovirals | 2 (6.5%) |
| Chemotherapy | 1 (3.2%) |
| **Other treatments** | |
| Proton pump inhibitors | 30 (96.8%) |
| Enteral nutrition | 23 (74.2%) |
| Opioids | 30 (96.8%) |
| **Motive for admission** | |
| Acute respiratory distress syndrome | 8 (25.8%) |
| Sepsis | 6 (19.4%) |
| Coma | 4 (12.9%) |
| Other | 13 (41.9%) |
| **Severity score at admission** | |
| SAPS II | 52 (25–106) |
| SOFA | 8 (1–16) |
| **Antibiotics within 21 day before inclusion** | |
| No | 17 (54.8%) |
| Yes | 14 (45.2%) |
| Amoxicillin + clavulanic acid | 4 (12.9%) |
| Amoxicillin | 5 (16.1%) |
| Third generation cephalosporin | 7 (22.3%) |
| Fourth generation cephalosporin | 2 (6.5%) |
| Piperacillin + tazobactam | 1 (3.2%) |
| Carbapenem | 1 (3.2%) |
| Fluoroquinolone | 3 (9.7%) |
| Amidazole | 2 (6.5%) |
| Aminoglycoside | 8 (25.8%) |
| Glycopeptide | 3 (9.7%) |
| Duration of antibiotic treatment (days) | 4 (1–27) |

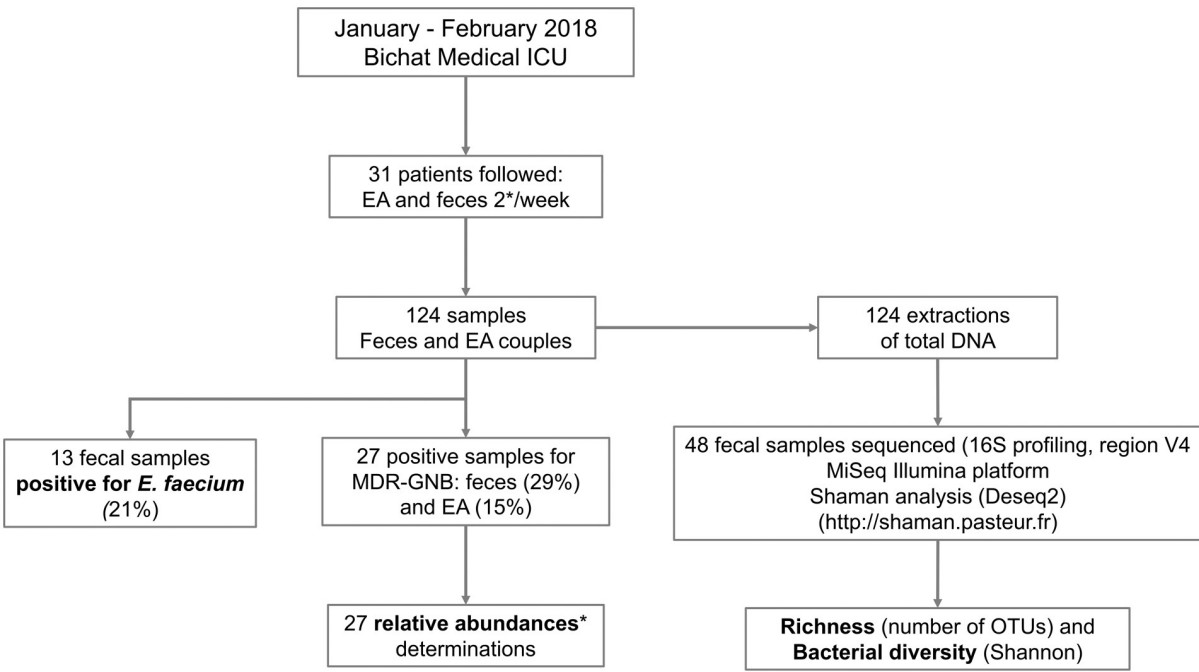

**Fig 1. Flow-chart of the study.** EA: endotracheal aspiration; MDR-GNB: Multi-Drug Resistant Gram-negative bacilli; Relative abundance: quantity of MDR-GNB / quantity of total GNB.

correlation test, p = 0.43) or richness (Pearson correlation test, p = 0.21) of the intestinal microbiota and the MDR-GNB intestinal relative abundance (Fig 2 panels 2C and 2D).

### *E. faecium* and yeasts intestinal colonization and composition of intestinal microbiota

However, we observed a significant link between intestinal colonization with *E. faecium* and the composition of the intestinal microbiota: the diversity (Student test, p = 0.04) and richness (Student test, p<0.001) of the intestinal microbiota were significantly lower in patients with *E. faecium* intestinal carriage (Fig 3 panels A and B). There was also a significant decrease in the diversity (Pearson correlation test, p<0.001) and richness (Pearson correlation test, p<0.001) and of the intestinal microbiota when the relative intestinal abundance of reads assigned to the *Enterococcus* genus increased (Fig 3 panels C and D). When considering only one sample per patient (the first collected), a significant linked remained between the relative abundance of reads assigned to the *Enterococcus* genus and diversity (Pearson correlation test, p<0.001) but not with richness (Pearson correlation test, p = 0.1) (S2 Fig in S1 File). Conversely, there was no significant difference between intestinal colonization by yeasts and diversity (Student test p = 0.09) or richness (p = 0.2) of the intestinal microbiota (S3 Fig in S1 File).

### Relationship between intestinal and endotracheal colonization with MDR-GNB

Fecal MDR-GNB colonization was high with an average intestinal concentration of 9.4 (per gram of feces expressed in log 10) (median 10.3 [7.2; 10.6]). In the trachea, the mean MDR-GNB concentration was 5.5 (median of 6.0 [4.6; 7.6]). Four patients (B, N, O, R) were found to be colonized at inclusion by the same MDR-GNB in the feces and the trachea. They

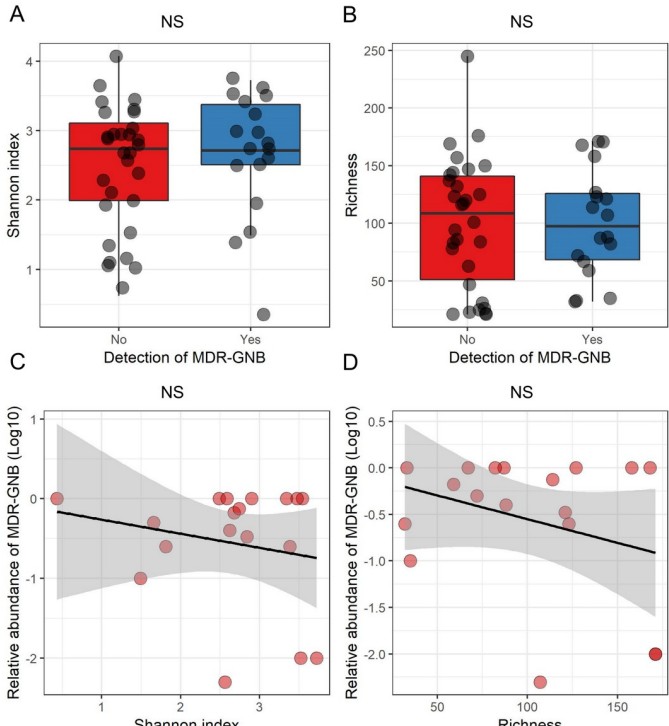

**Fig 2. MDR-GNB intestinal colonization and relative abundance and composition of the intestinal microbiota.**
MDR-GNB: Multidrug-resistant Gram-negative bacilli. Panels A and B: boxplot superimposed by dot-plot of Shannon diversity index (A) and richness (B) according to the detection by culture of MDR-GNB (n = 48 samples). Panels C and D: Dot-plot of the MDR-GNB intestinal relative abundance (in Log10), Shannon diversity index (C) and richness (D) (n = 18 samples). The shaded grey area depicts the 95% confidence interval around the black line. NS = not significant (panels A and B: Student test; panels C and D: Pearson correlation test).

had a higher intestinal relative abundance (Student test p = 0.02) of MDR GNB than those who did not, but not a higher concentration of MDR-GNB (Student test p = 0.08) (S4 Fig in S1 File).

## Discussion

The main result of this study is that we did not observe a link between the MDR-GNB intestinal RA and the richness or diversity of the intestinal microbiota of ICU patients. However, we did observe this link with enterococci. Indeed, there was a significant decrease in the richness and diversity of the intestinal microbiota parallel to the presence of *E. faecium* in the intestinal microbiota by culture and a significant increase in *Enterococcus* sp. relative abundance after 16S RNA sequencing. Enterococci have been showed to become dominant in the intestinal microbiota of ICU patients [19, 20] In the study by Freedberg *et al* in 2018 [21] intestinal colonization by an *E. faecium* and domination of the microbiota by enterococci as assessed by 16S rDNA sequencing at admission in the ICU were independent risk factors for 30-day mortality and for occurrence of infections with all germs. The relative abundance of *Enterococcus* sp. was also found to be associated to death in the study of Agudelo-Ochoa [22]. In the study of Lankelma et al., it was suggested that a higher intestinal diversity was associated to survival, but the mortality at D90 did not differ according to the intestinal diversity [17]. Our results show that the loss of richness and diversity of the intestinal microbiota when it is dominated by enterococci (including *E. faecium*) makes it a potential surrogate marker for intestinal

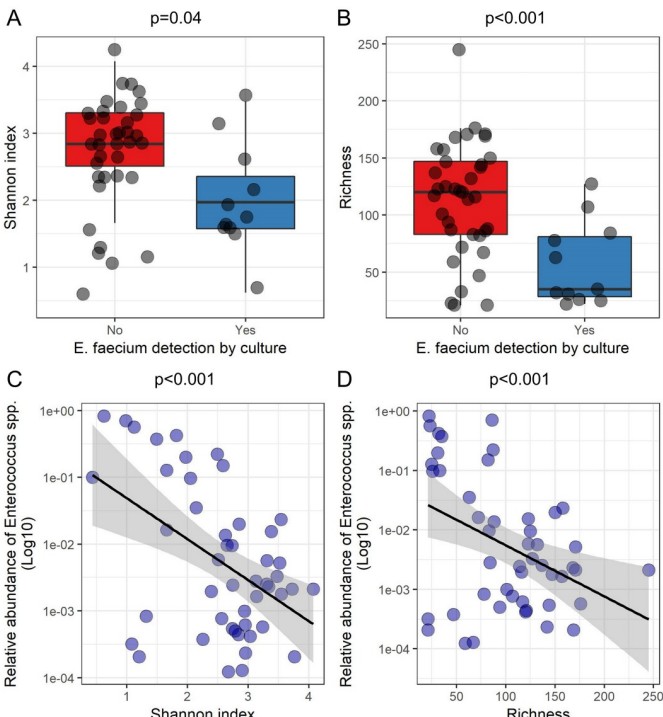

**Fig 3. *Enterococcus* spp. and *Enterococcus faecium* intestinal colonization according to the composition of the intestinal microbiota (16S profiling results from 11 samples containing *E. faecium*).** Panels A and B: boxplot superimposed by dot-plot of Shannon diversity index (A) and richness (B) according to the detection by culture of *E. faecium*. Panels C and D: Dot-plot of the relative abundance of reads assigned to *Enterococcus* spp. (in Log10), Shannon diversity index (C) and richness (D) (n = 48 samples). The shaded grey area depicts the 95% confidence interval around the black line. Panels A and B: Student test; panels C and D: Pearson correlation test.

dysbiosis. We did not observe the same link with intestinal colonization by yeasts detected in culture, but we could not to test the abundance of yeasts by sequencing since they are not spanned by the 16S profiling method.

Our study has limitations, though. This was a pilot study designed for assessing the connection between quantitative cultures and the global composition of the microbiota and the number of samples analysed in this regard may not have been sufficient to find a connection between the quantities of MDR-GNB and the composition of the intestinal microbiota. In addition, some samples come from the same patient and can potentially be very similar and therefore not independent, despite the use of microbiota-perturbing drugs between samples. Still, the linked between the intestinal relative abundance of reads assigned to the *Enterococcus* genus and the diversity remained significant when only one sample per patient was considered. Besides, we studied the intestinal microbiota composition and the MDR-GNB intestinal RA on fresh fecal samples. Collecting fresh spontaneous stool from resuscitation patients who are most often in functional occlusion was proven challenging and may not apply for large-scale studies in ICU. This should be overcome by the use of rectal swabs that can be collected more easily and at chosen times, but then the determination of the intestinal concentrations of bacteria may be compromised by the high variability of the fecal material collected by the swab. Last, we did not identify the yeasts species so that we were unable to test for species-specific associations.

In conclusion, we found no link between the MDR-GNB intestinal relative abundance or the MDR-GNB intestinal colonization and composition of intestinal microbiota. However,

this link was found with *Enterococcus* genus. Indeed, a significantly lower diversity and rich-ness of the intestinal microbiota was observed in patients colonized with *E. faecium* as well as when *Enterococcus* RA increased. *Enterococcus* seems to be an intestinal dysbiosis marker to be further explored.

## Supporting information

**S1 File.**
(DOCX)

**S1 Table. Characteristics of the patients included in this study.** SAPS: Simplified Acute. Physiological Score; S1-S8: dates of sampling; Antibiotic-21: antibiotic taken within 21 days before the inclusion. ATB: antibiotic.
(XLSX)

**S2 Table. Characteristics of the samples considered in this study.** ESBL: extended-spectrum beta-lactamase; hAmpC-Enterobacterales: high-level AmpC producing—Enterobacterales; CAZ-R: resistant to ceftazidime; MDR-GNB; multidrug-resistant Gram-negative bacilli; CFU: colony-forming unit.
(XLSX)

**S3 Table. Taxonomy of the operational taxonomic units (OTU) found in the fecal samples.** NA: not available.
(XLSX)

## Acknowledgments

We are grateful to the paramedical team of the medical ICU of the Bichat-Claude Bernard hos-pital for their assistance in collecting the samples. We also thank Amine Ghozlane (Institut Pasteur, Paris, France) for his technical support on Shaman and Marie Petitjean (IAME Research Center, Paris, France) for bioinformatic assistance.

## Author Contributions

**Conceptualization:** Laurence Armand-Lefèvre, Jean-François Timsit, Etienne Ruppé.

**Data curation:** Etienne Ruppé.

**Formal analysis:** Candice Fontaine, Anissa Nazimoudine.

**Investigation:** Candice Fontaine, Mélanie Magnan, Anissa Nazimoudine.

**Methodology:** Candice Fontaine, Etienne Ruppé.

**Resources:** Mélanie Magnan.

**Software:** Mélanie Magnan.

**Supervision:** Laurence Armand-Lefèvre, Etienne Ruppé.

**Validation:** Jean-François Timsit, Etienne Ruppé.

**Writing – original draft:** Candice Fontaine.

**Writing – review & editing:** Laurence Armand-Lefèvre, Jean-François Timsit, Etienne Ruppé.

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
