## [Decision Letter · Decision Letter 0]

16 Jun 2020

PONE-D-20-14488

Relationship between the composition of the digestive microbiota and the concentrations of opportunistic pathogens in intensive care patients.

PLOS ONE

Dear Dr. Ruppé,

Thank you for submitting your manuscript to PLOS ONE. After careful consideration, we feel that it has merit but does not fully meet PLOS ONE’s publication criteria as it currently stands. Therefore, we invite you to submit a revised version of the manuscript that addresses the points raised during the review process.

Please consider the comments of the reviewers to improve the manuscript.

We look forward to receiving your revised manuscript.

Kind regards,

Axel Cloeckaert

Academic Editor

PLOS ONE

Journal Requirements:

2. Please provide additional details regarding participant consent. In the ethics statement in the Methods and online submission information, please ensure that you have specified (1) whether consent was informed and (2) what type you obtained (for instance, written or verbal, and if verbal, how it was documented and witnessed). If the need for consent was waived by the ethics committee, please include this information.

3. Thank you for stating the following in the Funding Sources Section of your manuscript:

"This work was partially supported by the “Fondation pour la Recherche Médicale” (Equipe FRM 2016,

grant number DEQ20161136698)."

4. Thank you for stating in your Funding Section:

""This work was partially supported by the “Fondation pour la Recherche Médicale” (Equipe FRM 2016,

grant number DEQ20161136698).""

Reviewers' comments:

Reviewer's Responses to Questions

**Comments to the Author**

1. Is the manuscript technically sound, and do the data support the conclusions?

Reviewer #1: Yes

Reviewer #2: Partly

2. Has the statistical analysis been performed appropriately and rigorously? 

Reviewer #1: Yes

Reviewer #2: I Don't Know

3. Have the authors made all data underlying the findings in their manuscript fully available?

Reviewer #1: No

Reviewer #2: No

4. Is the manuscript presented in an intelligible fashion and written in standard English?

Reviewer #1: Yes

Reviewer #2: No

5. Review Comments to the Author

Reviewer #1: This was a retrospective single-center cohort study which included 31 patients who were mechanically ventilated for 3 or more days using a convenience sampling approach in medical ICU patients. Patients were sampled (stool and endotracheal aspirates (EA)) multiple times so there were 62 samples for 31 patients. Bacteria were cultured and 16S sequencing was performed. The main study goals were (1) to assess for a relationship between the EA samples and the fecal samples in terms of MDR Gram-negative bacteria and (2) assess for a relationship between overall bacterial diversity based on sequencing in those with versus those without MDR colonization (in both stool and EA samples). They found that there was evidence of an association between MDR GN content in the stool and EA samples. When Enteroccci were present, but not based on MDR GN colonization, there was an association with lack of microbial diversity in the samples. Overall, this is a useful contribution to the sparse literature on the ICU microbiome.

Major:

1. The apparent lack of a standardized sampling protocol is a major limitation. Rectal swabs may be a good way to overcome this limitation in the future, and the advantages of rectal swabs in terms of ease of collection and ability to time collections almost certainly outweigh the issues related to sample heterogeneity that the authors mention. This limitation cannot be overcome but it should be more plainly recognized. Also, a table detailing which samples were collected and when would be helpful. Were EAs always collected at the same time as fecal samples? If not, why not?

2. For the EA-stool comparison, test results and data should be presented as within-individual tests rather than as aggregate data.

Minor:

Title: The title makes it appear that only the gut microbiota were sampled. But endotracheal aspirates were also performed.

Abstract: The abstract does not report the correlation between the EA samples and the stool samples, arguably the most important finding of the study.

Methods:

1. More detail is needed related to the sampling. If everyone was sampled twice weekly, why weren’t there more samples? Perhaps because the stool samples were “missed” – but then why weren’t EA samples collected? If the EA samples were collected at different times than the stool samples, this is a significant limitation and needs to be recognized.

2. Explain the methods by which the dominant MDR GN were classified.

Results:

1. The sequencing data needs to be made freely available upon publication.

2. Table 1: Include the reason for admission. Also, less than 50% of patients received antibiotics which is low for the medical ICU. Why?

3. A table is needed showing the RA of the MDR GN organisms by organism type, stratified by stool vs EA. Lumping the MDR GN bacteria together risks misclassifying individuals. Ideally we would be presented for the EA-stool correlations within each individual, by organism.

4. P8 – “We considered samples positive for MDR-GNB…from a patient with no positive culture.” I don’t understand this sentence, it needs to be rephrased.

5. Figure 2 – these panels seem to contain fewer than 62 datapoints. Is this because data with zero RA has been excluded? The same goes for other figures (e.g., Supp 1). If not all data is being used, it should be recognized in the figure legend.

6. P9 – Stool and EA: were these the same organisms?

General: “fecal bacteria” would be more accurate than “intestinal bacteria” throughout the manuscript.

Reviewer #2: This manuscript provides an inadequate review of the literature on the microbial ecology of ICU patients, claiming that few such investigations have been conducted and citing just two previous studies. To provide an appropriate context for their work, they should dicusss and cite relevant previous studies, e.g. https://pubmed.ncbi.nlm.nih.gov/25249279,30561131,25249279,31526447,26715502,28803549,30526610,31924126

In addition, the research narrative is confusing and inadequate, as the authors lay the groundwork in the introduction for a study on multi-drug resistant Gram-negative bacilli, but then unexpectedly include Enterococci and yeast-like fungi in their study without providing a justification for this. The description of the fungal isolates is inadequate, as is the failure to perform susceptibility tests on enterococci.

Additional points:

They mention MDR Acinetobacter in the Introduction, but not anywhere else. It is unclear whether their culture methods are capable of detecting this organism but it simply was absent from their set of patients or whether their methods would not have found it. It is an important MDR GMB.

Sequence data have not been submited to and/or released by the ENA at the time of this review.

Sequencing of V4 alone is unable to provide resolution down to species level. This should be acknowleged as a deficit of the study.

Results of bioinformatics analyses not presented (e.g. OTU tables). No tabulation of samples, patients and cultured isolates.

The Discussion section should cite and discuss all the other studies that have reported enterococcal blooms in critical ill patients.

6. PLOS authors have the option to publish the peer review history of their article (what does this mean?). If published, this will include your full peer review and any attached files.

Reviewer #1: No

Reviewer #2: No

---

## [Author Response · Author response to Decision Letter 0]

20 Jul 2020

PONE-D-20-14488

Relationship between the composition of the digestive microbiota and the concentrations of opportunistic pathogens in intensive care patients.

Editor

Journal Requirements:

The style of the references has been updated to the Plos One format. 

 2. Please provide additional details regarding participant consent. In the ethics statement in the Methods and online submission information, please ensure that you have specified (1) whether consent was informed and (2) what type you obtained (for instance, written or verbal, and if verbal, how it was documented and witnessed). If the need for consent was waived by the ethics committee, please include this information.

The signed informed consent was not required by the Ethics Committee as the study was considered to be out of the scope of research performed on Humans. We added a sentence L112-113. 

 3. Thank you for stating the following in the Funding Sources Section of your manuscript:

"This work was partially supported by the “Fondation pour la Recherche Médicale” (Equipe FRM 2016, grant number DEQ20161136698)." We note that you have provided funding information that is not currently declared in your Funding Statement. However, funding information should not appear in the Acknowledgments section or other areas of your manuscript. We will only publish funding information present in the Funding Statement section of the online submission form.

The funding statement was removed from the text, and we modified the statement during the online re-submission. 

4. Thank you for stating in your Funding Section:

""This work was partially supported by the “Fondation pour la Recherche Médicale” (Equipe FRM 2016,grant number DEQ20161136698)."" Please provide an amended statement that declares *all* the funding or sources of support (whether external or internal to your organization) received during this study, as detailed online in our guide for authors at http://journals.plos.org/plosone/s/submit-now. Please also include the statement “There was no additional external funding received for this study.” in your updated Funding Statement.

This was added in the cover letter. 

The reads have been deposited at the NCBI SRA (access number PRJNA641109). 

The captions were added at the end of the manuscript.

Reviewer #1: This was a retrospective single-center cohort study which included 31 patients who were mechanically ventilated for 3 or more days using a convenience sampling approach in medical ICU patients. Patients were sampled (stool and endotracheal aspirates (EA)) multiple times so there were 62 samples for 31 patients. Bacteria were cultured and 16S sequencing was performed. The main study goals were (1) to assess for a relationship between the EA samples and the fecal samples in terms of MDR Gram-negative bacteria and (2) assess for a relationship between overall bacterial diversity based on sequencing in those with versus those without MDR colonization (in both stool and EA samples). They found that there was evidence of an association between MDR GN content in the stool and EA samples. When Enteroccci were present, but not based on MDR GN colonization, there was an association with lack of microbial diversity in the samples. Overall, this is a useful contribution to the sparse literature on the ICU microbiome.

Major:

1. The apparent lack of a standardized sampling protocol is a major limitation. Rectal swabs may be a good way to overcome this limitation in the future, and the advantages of rectal swabs in terms of ease of collection and ability to time collections almost certainly outweigh the issues related to sample heterogeneity that the authors mention. This limitation cannot be overcome but it should be more plainly recognized. 

This study was a pilot study aiming at setting grounds for wider studies. One of the endpoints was the feasibility of working with feces. We anticipated issues in the sampling scheme and this proved indeed difficult to obtain fecal samples from ICU patients. We have now started on working with fecal swabs, but in many cases the bacterial biomass and DNA yield are too low to be further used with molecular or even culture methods. In this regard, we may agree that fecal swabs could more standardized than fecal plain samples in a way they could be obtained at given times. However, we assume that they lack reproducibility in terms of fecal material they can recover. The issue of the sample type was discussed and completed in the revised version (L271-275).

Also, a table detailing which samples were collected and when would be helpful. 

We added a supplementary table for patients (Supplementary Table 1) and another one for samples (Supplementary Table 2). 

Were EAs always collected at the same time as fecal samples? If not, why not?

Yes, they were. When a patient had passed a stool, an EA was systematically taken (we added a specification L116). 

2. For the EA-stool comparison, test results and data should be presented as within-individual tests rather than as aggregate data.

We agree that ideally we should have compared the intestinal MDR-GNB concentrations and relative abundance between the first sample concomitant to the detection of MDR-GNB in EA to the previous one, but this would require that we would have several samples per patient. MDR-GNB were detected in 9 EA samples from 5 patients. In one (patient A), we found a S. maltophilia in EA but however not in the feces. In the other four patients (B, N, O and R) the MDR-GNB was found concomitantly in the EA and the feces from the first sample. Hence we could not compare the intestinal MDR-GNB concentrations and EA to another previous sample where the MDR-GNB would putatively not have been found yet in EA. Consequently, we performed a test on aggregate data. 

Minor:

Title: The title makes it appear that only the gut microbiota were sampled. But endotracheal aspirates were also performed.

We agree and changed the title to:

“Relationship between the composition of the intestinal microbiota and the tracheal and intestinal colonization by opportunistic pathogens in intensive care patients”.

Abstract: The abstract does not report the correlation between the EA samples and the stool samples, arguably the most important finding of the study.

We emphasized the intestinal microbiota as our main objective was to assess the link between the composition of the intestinal microbiota (richness and diversity) and the carriage (quantitative and qualitative) of MDR-GNB (this was clarified at the end of the introduction (L99-101). We hypothesized we would obtain a correlation between richness/diversity and intestinal concentrations/relative abundance of MDR-GNB, which we eventually did not. The connection with EA was a secondary objective. We actually published a paper in which the main objective was to connect the concentrations/relative abundances of ESBL-E in EA and fecal swabs and the occurrence of infections caused by ESBL-E (Andremont O, Armand-Lefevre L, Dupuis C, de Montmollin E, Ruckly S, Lucet J-C, et al. Semi-quantitative cultures of throat and rectal swabs are efficient tests to predict ESBL-Enterobacterales ventilator-associated pneumonia in mechanically ventilated ESBL carriers. Intensive Care Med. 2020). 

Methods:

1. More detail is needed related to the sampling. If everyone was sampled twice weekly, why weren’t there more samples? Perhaps because the stool samples were “missed” – but then why weren’t EA samples collected? If the EA samples were collected at different times than the stool samples, this is a significant limitation and needs to be recognized.

As previously mentioned, the EA were taken the same day the feces were obtained. The samples are now detailed in the Supplementary Tables 1 and 2. In many instances, the sampling schedule was stopped in case of extubation, discharge, death or the absence of feces. When no stool was passed, no EA was collected. 

2. Explain the methods by which the dominant MDR GN were classified.

When a patient carried more than one MDR-GNB in feces or in EA, the MDR-GNB RA was the RA of the total MDR-GNB We were not able to provide as many RA as there was MDR-GNB. This was specified L145. However, the presence of two MDR-GNB in a sample only occurred twice (samples K_FEC_1 and AE_FEC_2). 

Results:

1. The sequencing data needs to be made freely available upon publication.

The reads have now been deposited at the NCBI SRA (access number PRJNA641109). 

2. Table 1: Include the reason for admission. 

The mains reasons for admission was added in the table 1. Furthermore, we detailed the individual reasons for admission in the Supplementary Table 1. 

Also, less than 50% of patients received antibiotics which is low for the medical ICU. Why?

This was a typo for which we apologize. In the table 1 this should read “antibiotics 21 days before the inclusion”. This was corrected. The details of antibiotic therapies are now detailed in the Supplementary Table 1. 

3. A table is needed showing the RA of the MDR GN organisms by organism type, stratified by stool vs EA. Lumping the MDR GN bacteria together risks misclassifying individuals. Ideally we would be presented for the EA-stool correlations within each individual, by organism.

The detailed microbiological results are now included in the Supplementary Table 2. 

4. P8 – “We considered samples positive for MDR-GNB…from a patient with no positive culture.” I don’t understand this sentence, it needs to be rephrased.

The sentence has been rephrased (L205-210). Furthermore, the motivation for sequencing was specified in the Supplementary Table 2. 

5. Figure 2 – these panels seem to contain fewer than 62 datapoints. Is this because data with zero RA has been excluded? The same goes for other figures (e.g., Supp 1). If not all data is being used, it should be recognized in the figure legend.

The panels depicting the detection of MDR-GNB (A and B) contain 48 samples. Not all fecal samples were sent to 16S profiling, only 48 were (L205-210 and Supplementary Table 2). For clarification matters, we specified the number of samples depicted in the graphics in the legends. 

6. P9 – Stool and EA: were these the same organisms?

Yes, the same species was found in the feces and the trachea. The paragraph was rephrased. (This was specified in the manuscript (L238 ). 

General: “fecal bacteria” would be more accurate than “intestinal bacteria” throughout the manuscript.

Does the reviewer refer to intestinal microbiota? We could not find “intestinal bacteria” in the manuscript. We changed intestinal MDR-GNB for fecal MDR-GNB (L44 and L238). 

Reviewer #2: This manuscript provides an inadequate review of the literature on the microbial ecology of ICU patients, claiming that few such investigations have been conducted and citing just two previous studies. To provide an appropriate context for their work, they should dicusss and cite relevant previous studies, e.g. https://pubmed.ncbi.nlm.nih.gov/25249279,30561131,25249279,31526447,26715502,28803549,30526610,31924126

In addition, the research narrative is confusing and inadequate, as the authors lay the groundwork in the introduction for a study on multi-drug resistant Gram-negative bacilli, but then unexpectedly include Enterococci and yeast-like fungi in their study without providing a justification for this. 

We completed the introduction with the references provided by the Reviewer 2 (L88-101). The justification of studying the link between the composition of the microbiota and enterococci/yeasts is now better justified by the observations reporting that Enterococcus spp. and yeasts often become dominant in the gut of critically-ill patients, and that Enterococcus spp. may also be a marker of worse outcome. 

The description of the fungal isolates is inadequate, as is the failure to perform susceptibility tests on enterococci.

At this stage, we did not hypothesize that yeasts would be differentially associated to the composition of the intestinal microbiota with respect to their species. Given the small number of patients, testing such association would not have been possible. This was added in the limitation section of the discussion. 

Likewise, we did not consider the antibiotic susceptibility profiles of cultured E. faecium in this study as it was not assumed to play a role in the link between their detection and the composition of the intestinal microbiota. 

Additional points:

They mention MDR Acinetobacter in the Introduction, but not anywhere else. It is unclear whether their culture methods are capable of detecting this organism but it simply was absent from their set of patients or whether their methods would not have found it. It is an important MDR GMB.

MDR A. baumannii was sought by culturing on the ChromID-ESBL medium, but none of the samples was positive. For clarification, we removed the reference to A. baumannii in the introduction. 

Sequence data have not been submitted to and/or released by the ENA at the time of this review.

The reads have been deposited at the NCBI SRA (access number PRJNA641109). 

Sequencing of V4 alone is unable to provide resolution down to species level. This should be acknowleged as a deficit of the study.

We agree that sequencing the V4 alone, and more generally, 16S profiling does not provide a deep taxonomic resolution. In this perspective, several studies are somewhat frustrating in that they report relevant OTUs that are not precisely identified. We did not aim at identifying specific taxa associated to the intestinal carriage or concentrations of MDR-GNB or Enterococcus and our sample schedule was not designed for this purpose. We did not report any result at the species level but only report overall OTU-level richness and diversity, and the relative abundance of Enterococcus spp. Hence, we assume that sequencing the V4 region was sufficient to cluster OTUs and estimate richness and diversity. 

Results of bioinformatics analyses not presented (e.g. OTU tables). No tabulation of samples, patients and cultured isolates.

We added a table (Supplementary Table 3) with the OTU count and taxonomy. Besides, we added two tables with patients (Supplementary Table 1) and samples (Supplementary Table 2) details. 

The Discussion section should cite and discuss all the other studies that have reported enterococcal blooms in critical ill patients.

We completed the discussion and included some of the references suggested by the Reviewer 2 (L250-251 and L254-255)

---

## [Editor Report · Decision Letter 1]

23 Jul 2020

Relationship between the composition of the intestinal microbiota and the tracheal and intestinal colonization by opportunistic pathogens in intensive care patients.

PONE-D-20-14488R1

Dear Dr. Ruppé,

We’re pleased to inform you that your manuscript has been judged scientifically suitable for publication and will be formally accepted for publication once it meets all outstanding technical requirements.

Kind regards,

Axel Cloeckaert

Academic Editor

PLOS ONE
---

## [Editor Report · Acceptance letter]

20 Aug 2020

PONE-D-20-14488R1 

Relationship between the composition of the intestinal microbiota and the tracheal and intestinal colonization by opportunistic pathogens in intensive care patients. 

Dear Dr. Ruppé:

I'm pleased to inform you that your manuscript has been deemed suitable for publication in PLOS ONE. Congratulations! Your manuscript is now with our production department. 

Kind regards, 

on behalf of

Dr. Axel Cloeckaert 

Academic Editor

PLOS ONE